# Associations between culture of health and employee engagement in social enterprises: A cross-sectional study

Patrick Nekula[☯], Clemens Koob[ID]*[☯]

Department of Health and Nursing, Catholic University of Applied Sciences Munich, Munich, Bavaria, Germany

☯ These authors contributed equally to this work.
* clemens.koob@ksh-m.de

## Abstract

### Introduction

The aging of staff and skill shortage are major challenges for social enterprises. Nurturing a workplace culture of health and fostering employee engagement could be starting points to combat these challenges. The associations between these two factors have received comparatively little attention from the scientific community, in particular with regard to social enterprises. Hence, this study aims to examine those associations, drawing on the job demands-resources theory and the social-ecological workplace culture of health model. It is hypothesized that employees' self-rated health acts as a mediator in the relationship between culture of health and employee engagement and that health as personal value works as a moderator.

### Method

The study used the Workplace Culture of Health scale to measure culture of health in social enterprises and UWES-9 to assess employee engagement. Data was collected administering a quantitative online survey among employees of social enterprises in Germany. The dataset for analyses comprised N = 172 employees in total. Data analyses included Pearson's correlations, regression analysis, as well as mediation, moderation and moderated mediation analyses.

### Results

Culture of health is a predictor of employee engagement in social enterprises. The analyses demonstrate a moderate association between culture of health and employee engagement. Indications were found that employees' self-rated health acts as a mediator and that health as personal value acts as a moderator between culture of health and employee engagement in social enterprises.

### Discussion

This study suggests that fostering a culture of health in social enterprises does not only have a positive effect on employee health, but also on employee engagement. This applies

**Data Availability Statement:** All relevant data are within the paper and its Supporting Information files.

**Funding:** The authors received no specific funding for this work.

**Competing interests:** The authors have declared that no competing interests exist.

in particular when employees attribute great value to their health, which is to be expected even more in future. Hence, nurturing a culture of health becomes a pivotal management task in social enterprises. Moreover, a comprehensive assessment of the benefits of health promotion programs in social enterprises should not only consider their health-related outcomes, but also factor in their impact on employee engagement.

## Introduction

The aging of society is a well-documented phenomenon in developed countries [1] which has a tremendous impact on organizations as it leads to a constantly aging workforce [2]. This poses major challenges for organizations in general and social enterprises, i.e. organizations that primarily provide social services [3, 4], in particular [5]. The rising average age of employees is generally associated with increasing age-related health problems, implying a decreasing ability to work and a higher risk of prolonged sick leaves [6, 7]. These factors are in turn associated with, e.g., losses in productivity and performance and substantial cost burdens [6]. Another threat to organizations in the social work sector that is at least partially caused by a society-wide aging workforce is the continuously growing shortage of skilled professionals [8]. Taking these developments together, social enterprises are required to preserve their employees' capacity to work, i.e. prevent or counter age-related or other kinds of illnesses and diseases. In addition, employee fluctuation needs to be prevented, while recruiting and motivating skilled professionals, to combat the threat of skill shortage.

Nurturing a workplace culture of health could be one starting point to meet these challenges. According to Schein's conceptualization organizational culture is being shaped by a continuous learning process and comprises three levels: artifacts, values and assumptions [9]. A workplace culture of health denotes an organizational culture that prioritizes and promotes employee health and well-being at all of these three levels. In the case of a pronounced culture of health, health promotion programs may be an integral part of business on the artifact level, while on the level of values, health may be an essential aspect of guiding principles and strategies, positively influencing managers' and employees' assumptions regarding health and well-being at the deepest level of organizational culture [10, 11].

Another starting point for tackling the aforementioned challenges could be fostering employee engagement. This approach was generally found to be effective in prior studies, e.g. in attenuating turnover-intentions and thereby reducing employee fluctuation, or in improving the mental and physical health of employees [12–14]. Employee engagement is usually defined as 'a positive, fulfilling, work-related state of mind that is characterized by vigor, dedication, and absorption' [15]. This definition conceptualizes engagement as comprising three dimensions. Whereas vigor describes high levels of energy and resilience regarding one's work, dedication refers to work-related feelings of pride and significance and absorption denotes a particularly concentrated, flow-like working state during which it seems hard to disconnect from working [15].

While previous studies suggested that both promoting a workplace culture of health and encouraging employee engagement are potential remedies for the challenges social enterprises are facing, there are no studies yet about how these factors are related. Greater understanding of the relationship between these two factors may help extend research on workplace culture of health by exploring employee engagement as potential outcome, and may at the same time add to research on employee engagement by introducing health culture as possible antecedent. In addition, a more precise knowledge of the relationship between health culture and work

engagement may help to further professionalize the management of social enterprises, which scholars have called for [16, 17]. Hence, this study aims to investigate the associations between workplace culture of health and employee engagement in the domain of social enterprises.

## Theoretical background

**Job demands-resources theory.** To investigate the associations between a workplace culture of health and employee engagement, this study first draws on the job demands-resources (JD-R) theory. This model of occupational well-being is commonly used by researchers to examine potential antecedents of employee engagement [13]. As depicted in Fig 1, the model differentiates between two processes.

If job demands, which are defined as 'physical, social, or organizational aspects of the job, that require sustained physical or mental effort and are therefore associated with certain physiological and psychological costs' [18], are too high or cannot be met, they can lead to strain or exhaustion which in turn can lead to negative health outcomes [13, 19, 20]. In the JD-R model, this is termed health impairment process. The other process of the model is based on job resources. Job resources are defined as 'physical, psychological, social, or organizational aspects of the job that may do any of the following: (a) be functional in achieving work goals; (b) reduce job demands and the associated physiological and psychological costs; (c) stimulate personal growth and development' [18]. If adequate job resources are available to fulfill job demands, this can lead to engagement, which in turn can lead to positive performance outcomes. This is described as the motivational process of the JD-R model [13, 20]. It is important to add that, though not depicted in Fig 1, an interaction between job demands and job resources is assumed, since initiating motivation or health impairment processes is dependent on whether there are enough job resources available to meet the job demands, implying a relation between the two factors [21]. According to meta-analyses, there are further outcomes of high levels of employee engagement besides employees' task performance, such as increased levels of organizational citizenship behavior and reduced turnover intentions [22]. Referring to these potentially beneficial outcomes of employee engagement, it seems plausible that social enterprises can take measures to increase levels of employee engagement in order to combat current challenges they are facing, as stated above. Based on various research efforts, it is also assumed that personal resources play a role in the JD-R model, but due to the diversity of studies attributing varying roles to these personal resources, so far no broad consensus on their position in the model has been reached [12, 23].

Since the goal of the present study is to examine the association between employee engagement and a workplace culture of health, we will conceptualize the latter in the next section.

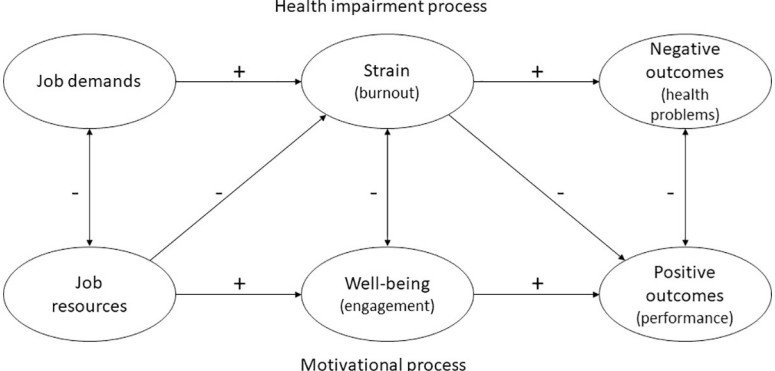

**Fig 1. Job demands-resources (JD-R) model [14].**

**Social-ecological workplace culture of health model.** In line with prior research [e.g. 11, 24] we conceptualize a health culture as an organizational culture that prioritizes and promotes employee health and well-being on all cultural levels. We further draw on the workplace culture of health theory proposed by Kwon and colleagues [25], being one of the most comprehensive concepts elaborated in this domain so far. This concept resorts to a social-ecological model suggesting that individual behaviors are influenced by both environmental and social factors. Other influencing factors are personal characteristics and interpersonal processes [26]. Therefore, these multidimensional factors and their interactions should be taken into account when analyzing health and health promotion [26, 27]. Based on this social-ecological model Kwon and colleagues [25] describe a workplace culture of health as a construct consisting of environmental and social factors in and of organizations, that can influence the (health-) behavior and health of individuals. The derived factors supporting workplace health and their definitions are depicted in Table 1.

Prior research demonstrated positive effects of a strong workplace culture of health on employee health and health-promotion-effectiveness [24, 28], but also suggested positive distal effects on employee [29] and stock performance [30, 31]. Referring to these potentially beneficial outcomes, it seems plausible to assume that promoting a workplace culture of health can be a legitimate strategy to counter the challenges social enterprises are confronted with, as outlined before. Hence, in what follows, we will discuss the possible associations between a workplace culture of health and employee engagement.

## Research hypotheses

**Workplace culture of health as antecedent of employee engagement.** Based on the theoretical building blocks outlined before, we propose health resources to be the linking pin between a workplace culture of health and employee engagement. Health resources can be defined as resources that prevent or weaken the effect of health-related stressors on individuals and in turn may have a positive impact on individuals' health and well-being, as well as recovery. In the context of work, examples for health resources on an individual level are professional health competencies or resilience, and examples on an organizational level are autonomy or learning opportunities [32]. As part of the JD-R model, health resources might

**Table 1. Definitions of the culture of health dimensions [25].**

| Dimension | Definition |
|---|---|
| Senior leadership | Expressed vision and resource allocation from the senior leaders indicating the employee's health is a priority for the organization |
| Policies and procedures | Alignment to support and accomplish vision in matters of health; and serves as a catalyst to allow employees to benefit from available resources |
| Programs | Initiatives and programs to support and improve employee health |
| Supervisor support | Encouragement, concern, and support from supervisors regarding a support for individual health and health promotion initiative |
| Coworker Support | Encouragement and support from peers regarding health |
| Role modeling | Other's practice of healthy behaviors or setting health as a priority; living evidence that certain achievements are possible |
| Mood | Employee attitudes, feelings, and perceptions that can influence motivation; mood can enhance or inhibit program participation |
| Values* | In general, values stem from the leaders of an organization, and are then cultivated among managers and employees. In a culture of health, employee health tends to be viewed as having intrinsic worth and is related to the organization's success. |

* Note: Values was added according to the latest revision of the workplace culture of health scale.

help to reduce job demands and foster engagement. Since a workplace culture of health aims to promote health resources, we expect a strong workplace culture of health to positively affect employee engagement in social enterprises.

Taking a closer look at the dimensions of a workplace culture of health should help to further unfold this proposed mechanism. By crafting a compelling health-related vision *senior leaders* could clearly emphasize the importance of promoting health in the organization and lead to more awareness for health [33], and thus contribute to an increase in employees' health resources. Organizational *policies and procedures* might work in a similar vein, allowing employees to benefit from available resources. An example would be catering policies focusing on educating employees in nutritional issues and increasing their health-related knowledge [34]. Health promotion *programs* refer to efforts to enrich health resources in organizations and are thus expected to be also positively related to employee engagement. Behavioral focused programs target individuals directly, e.g. through the means of company sport groups, and environmental-focused programs aim to change work conditions to promote employee health, e.g. through reducing overtime [32, 35]. *Social support* at work generally denotes the level of helping social interactions with management, supervisors, or coworkers at work [36], can be a way to access resources that are beyond those of individuals [20, 37] and is known to be positively related to employee engagement [38, 39]. We expect this positive relationship with engagement also to hold for health-related support, since supervisors' and coworkers' encouragement and concern in health matters are equivalent to an increase in an employee's health resources. Referring to Bandura's social-cognitive learning theory, individuals are able to adapt to their environment by observing and mimicking models surrounding them. In an organizational context, supervisors or colleagues could act as such *role models for healthy behaviors*, e.g. by dealing with stress in a sensible way, stimulating an employee's personal growth and development in health matters [40, 41]. This corresponds to an accrual of health resources and should therefore have a positive effect on employee engagement. A positive *mood* can also function as an individual's personal resource, since it refers to psychological 'aspects of the self that are generally associated with resiliency and that refer to the ability to control and impact one's environment successfully' [12], and it is therefore expected to be positively related to work engagement. If, e.g., an employee is convinced that the social enterprise she is working for has her best health interests at heart, it can be expected that she is more persistent in dealing with work demands, and engagement is more likely to occur. Organizational *values* encourage certain practices, give meaning to actions and are apt to increase employees' connectedness to their work environment [42]. Within the JD-R framework, values can function as job resources that can help achieving work goals, reducing job demands or stimulating personal growth and development. It can be expected that this also applies to health-related organizational values. If health is considered a core value, it means an increase in health-related resources for employees, since it enables them to take their own health into account and act in accordance with culture at the same time, which should contribute positively to employee engagement.

Taken together, we expect:

*H1*: *A workplace culture of health is positively related to employee engagement in social enterprises.*

**Self-rated health as a potential mediator in the relation between culture of health and employee engagement.** The central goal of the present study is to examine the relationship between a workplace culture of health and employee engagement. To do so, it is necessary to dig even deeper into the potential effects of a workplace culture of health. The social-ecological workplace culture of health model implies that a strong health culture is apt to positively influence employee's health-related perceptions, in other words, their self-rated health. Self-rated

health refers to a complex cognitive process including various influencing factors, ranging from personal aspects like physical activity, tobacco and alcohol use, diet or obesity, to the cultural environment as an influencing factor on individual's self-perception [43, 44]. Kwan and Marzec [45], e.g., have argued that a strong culture of health makes employees more interested in practicing healthy behaviors which affects self-rated health.

According to the JD-R theory, a positively rated health in turn could act as a personal resource and is hence supposed to increase employee engagement. Prior research by Brauchli and colleagues [46] adds support to this line of argumentation, in finding that employees' self-rated health can be an antecedent to employee engagement, as does a study from Persson and colleagues that demonstrated a positive relation between self-rated health and overall work experiences [47].

Consequently, it might be expected that employees' self-rated health mediates the relationship between a workplace culture of health and employee engagement. In addition to a direct positive effect on employee engagement, an established workplace culture of health could have a positive effect on employees' self-rated health, which in turn also could lead to a positive effect on employee engagement.

Therefore, the following hypothesis is proposed:

*H2: Self-rated health acts as mediator in the relationship between workplace culture of health and employee engagement in social enterprises.*

**Health as a personal value as a possible moderator of the association between workplace culture of health and employee engagement.** Previous research on organizational health culture has primarily purported positive effects of workplace culture of health initiatives. However, it is also known from past general research on cultural change, that the effects of change initiatives depend on the personal values of employees [48]. Prior studies have demonstrated that employees exposed to active efforts to change an organization's culture may feel pushed to act inauthentically [49, 50] in cases when the intended culture makes it difficult for them to be true to themselves and act in accordance with their personal values [51–53], leading to less acceptance and less energetic support of these efforts.

For the present study this means that we similarly expect that the effects of a culture of health on employee engagement will depend on the degree to which the culture allows employees to 'just be themselves', i.e. to act authentically. Following this perspective, employees that attribute great value to their health are likely able to act authentically under the conditions of a strong health culture, while employees that regard health as less important may find it not that easy to behave authentically. Hence, the former group of employees might experience stronger engagement uplifts with a more pronounced workplace culture of health than the latter group.

In other words, we expect the following:

*H3: Health as personal value acts as a moderator between workplace culture of health and employee engagement in social enterprises.*

## Method

### Study design and setting

To test the proposed hypotheses, a cross-sectional research design was chosen. For collecting primary data, we relied on a structured questionnaire and used measures from previous research. All questions were asked in German language. The questionnaire was hosted on an online platform (SoSciSurvey) to fulfil the European Union's data privacy rules. Data were collected in August and September 2019.

## Participants

The target group of the study were employees of social enterprises in Germany. More specifically, criteria for inclusion were that the employing organization primarily provided social services and that social workers or social pedagogues were part of the workforce. Participants themselves did not have to be social workers or social pedagogues, since other professionals (e.g., administrative workers, psychologists) also work in social enterprises. Moreover, participants needed to be employed at the current social enterprise for at least 6 months, since it typically takes this time to ensure sufficient organizational socialization [54] which is required for understanding and reporting on an organization's culture [9].

On the one hand, potential participants were directly contacted via social media using three well-known social work-related groups on Facebook. On the other hand, based on a systematic identification of social enterprises in different regions of each federal state in Germany, more than 150 organizations were contacted by phone and asked to support the study. Those who declared their willingness to support the investigation received an e-mail describing the study goals and procedure and were asked to forward the invitation for participation to employees.

## Measurement of main variables

*Employee engagement* was measured with the German version of the Utrecht Work Engagement Scale-9 (UWES-9). It is the short version of the UWES questionnaire consisting of three subscales (vigor, dedication, absorption) with three items each that are measured on a 7-point Likert type scale (ranging from never to always). In previous research, this scale showed high internal consistency and test-retest-reliability, as well as discriminant, convergent and construct-validity and therefore was deemed appropriate for this study [22, 55, 56].

*Workplace culture of health* was measured with the most recent version of the Workplace Culture of Health Scale developed by Kwon and colleagues [25]. In order to use the latest version, the corresponding author of the scale development study was directly contacted. This scale was chosen because it was the only scale that explicitly surveys workplace culture of health from an employee perspective. Furthermore, it was validated several times in prior research and showed signs of high internal consistency with evidence of convergent and discriminant validity [25, 57]. The present study used those 36 items from the scale that are aimed at measuring a culture of health (i.e., items proposed by Kwon and colleagues [25] that are not geared towards measuring a workplace culture of health, but, e.g., towards capturing details of corporate wellness programs, were not included). All workplace culture of health items were measured on a 6-point Likert type scale (ranging from strongly disagree to strongly agree). The items were divided into subscales representing leadership, policies and procedures, programs, supervisor support, coworker support, role modeling, values, and mood.

*Self-rated health* is frequently measured with a single item. A comparative study of three different single-item scales for measuring self-rated health concluded that all examined scales could be legitimately used to measure self-rated health [58]. Therefore, this study used one single-item measure that is commonly relied on in big scale surveys in Germany to measure self-rated health [59]. Study participants were asked to answer the question 'How is your general state of health?' using a 5-point Likert type scale (ranging from very bad to very good).

As in prior research in the workplace culture of health domain, *health as a personal value* was also measured using a single item. The item was drawn from the Workplace Culture of Health Scale [25]. Respondents were asked to state on a 6-point Likert type scale (ranging from strongly disagree to strongly agree) whether 'taking care of my health is a strong priority in my life'.

Within the scope of the study, all items drawn from the Workplace Culture of Health Scale had to be translated into German language. The translation process followed the guidelines for cross-cultural adaption of self-report measures [60]. The scale items were first forward translated from English to German language by a bilingual translator and then translated back into English language by another bilingual translator blind to the original version. Afterwards, the translations were reviewed, and adjustments were made where needed to achieve semantic, idiomatic, experiential and conceptual equivalence. Before conducting the study, a pretest of the questionnaire was carried out. A content-related pretest, with a focus on comprehensibility, clarity and appropriateness of translation of those items that were translated as part of the study, was carried out with 10 people from the target group in 2 phases and the feedbacks were taken into account. A technical pretest did not show any problems.

In addition to the above variables, *gender* (binary coded) and *age* (years) were recorded to be included in the analyses as potential controls, since both factors may significantly relate to the variables under investigation [e.g., 55], possibly leading to a 'mixing of effects'.

## Study size

We eliminated responses from the sample that failed to fit in the target group as described above. Furthermore, based on pretests, responses that took less than 5 minutes to complete were also eliminated from the sample, to counteract participants skimming over the questionnaire and not answering the questions seriously.

## Statistical analyses

We used regression analyses including mediation and moderation analyses for hypotheses testing. Statistical analyses were performed using IBM SPSS Statistics 26. Mediation and moderation analyses were carried out with Hayes' [61] PROCESS Macro version 3.5. The procedure comprised four steps: First, to evaluate the effect of a workplace culture of health on employee engagement, engagement was regressed on health culture. Second, the potential mediating effect of self-rated health was examined with regression analysis using Hayes' PROCESS Macro, model 4. Third, the possible moderating effect of health as personal value was examined with regression analysis using Hayes' PROCESS Macro, model 1. To finally jointly investigate the hypothesized effects of workplace culture of health, self-rated health and health as personal value on employee engagement, regression analysis using Hayes' PROCESS Macro, model 5, was carried out. A p-value of $< .05$ was considered significant.

## Ethical considerations

Before realizing the study, the University Ethics Review Board regulations indicated that a research ethics review was not necessary. Reasons for this decision are that the investigation does not include any manipulations or vulnerable groups, and participants were guaranteed that their data is treated anonymously. Moreover, the data has been collected in accordance with the EU General Data Protection Regulation. All participants provided informed consent by clicking on the link to start the study, participation was completely voluntary, and only data from participants were used who completed the study.

## Results

### Participant data

In total, data collection yielded 213 responses. After eliminating responses from the sample that failed to fit with the aforementioned inclusion criteria, the final sample for analyses

comprised N = 172 employees from social enterprises. The sample consisted of 120 (69.8%) female participants, 51 (29.7%) male participants and 1 (0.6%) person identifying as neither female nor male. The high proportion of women in the study sample is characteristic for employees in social enterprises in Germany [62]. The average age of the respondents was 38.0 years (SD 11.25), which is only slightly lower than the mean age of employees in the social sector according to a nationwide survey (41.6 years, [63]), and 132 (76.7%) of the respondents had a degree in social work or social pedagogy. In sum, the characteristics of respondents were in line with expectations.

### Descriptive statistics and correlations

Table 2 lists the means, standard deviations, Pearson's correlations, and Cronbach's alphas of the study variables.

Cronbach's alpha coefficients for the multi-item measures workplace culture of health and employee engagement were .95 and .94, exceeding the recommended minimum of .70, indicating a very good reliability [64].

In line with expectations, employee engagement related positively to workplace culture of health (r = .48, p < .01) and to self-rated health (r = .32, p < .01), with the correlation coefficients indicating moderate relations [65] between the variables. In addition, workplace culture of health showed weak correlations with self-rated health (r = .18, p < .05) and health as a personal value (r = .28, p < .01), as did self-rated health and health as a personal value (r = .27, p < .01).

### Hypothesis testing

To evaluate the effect of a workplace culture of health on employee engagement, regression analysis was used. Workplace culture of health explained a substantial proportion of variance in employee engagement ($R^2$ = .23, F(1, 170) = 51.49, p < .001). In Hypothesis 1, we expected that there would be a positive association between a workplace culture of health and employee engagement in social enterprises. The regression coefficient indicated that as we hypothesized, workplace culture of health was significantly and positively associated with engagement (b = .68, t(170) = 7.18, p < .001). Therefore, the data support Hypothesis 1.

With regard to Hypothesis 2, we predicted that self-rated health would act as mediator in the relationship between workplace culture of health and employee engagement in social enterprises. The potential mediating effect of self-rated health was examined using Hayes' PROCESS Macro, model 4. Results of the mediation analysis are presented in Fig 2.

Workplace culture of health was positively associated with self-rated health (a = .17, t(170) = 2.33, p < .05), which in turn was positively related to employee engagement (b = .35,

**Table 2. Means, standard deviations, correlations and Cronbach's alphas of study variables.**

| Variables | M (range) | SD | Items | 1 | 2 | 3 | 4 |
|---|---|---|---|---|---|---|---|
| 1. Culture of health | 4.01 (1–6) | .73 | 36 | *.95* | | | |
| 2. Engagement | 5.10 (1–7) | 1.04 | 9 | .48** | *.94* | | |
| 3. Self-rated health | 3.89 (1–5) | .71 | 1 | .18* | .32** | -- | |
| 4. Personal value health | 4.73 (1–6) | 1.01 | 1 | .28** | .12 | .27** | -- |

Notes

* p < .05

** p < .01

Cronbach's alphas for multi-item measures are in italics on the diagonal in the correlation matrix.

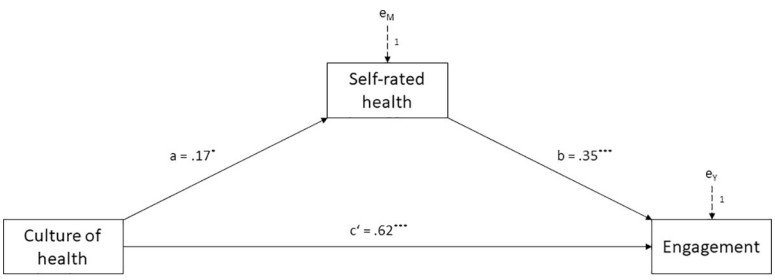

Notes: * p < .05, *** p < .001

**Fig 2. Mediation model.** Self-rated health mediating the effect of workplace culture of health on employee engagement.

t(169) = 3.66, p < .001). Significance of the indirect effect was examined using bootstrapping. As recommended by Hayes [61], 5'000 bootstrapped samples and a confidence interval of 95 percent were used. The analysis yielded a positive and significant indirect effect of workplace culture of health on employee engagement through self-rated health of a x b = .06 with a 95 percent confidence interval from .002 to .130. At the same time, the residual direct effect of workplace culture of health on employee engagement was also significant (c' = .62, t(169) = 6.67, p < .001). Therefore, self-rated health partially mediated the relation between a workplace culture of health and employee engagement. The model accounted for 29 percent of the variance in employee engagement ($R^2$ = .29, F(2,169) = 34.30, p <. 001). Thus, Hypothesis 2 cannot be rejected.

Hypothesis 3 predicted a moderating role of health as personal value between culture of health and employee engagement. The moderating effect of health as personal value was examined using Hayes' PROCESS Macro, model 1. Following recommendations by Hayes [61], we mean-centered the predictor (i.e. culture of health) and the moderator (i.e. health as personal value) prior to analysis to aid in interpretation. Results of the moderation analysis are presented in Fig 3.

The analysis yielded a significant model accounting for 27 percent of the variance in employee engagement ($R^2$ = .27, F(3,167) = 20.91, p < .001). We found a significant interaction between workplace culture of health and health as personal value on employee

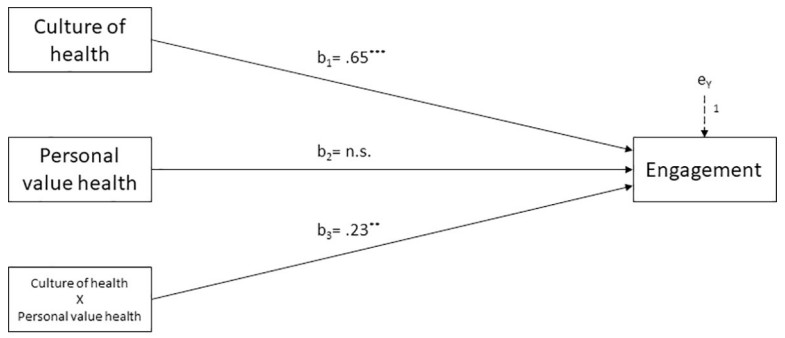

Notes: ** p < .01, *** p < .001; n.s. = not significant

**Fig 3. Moderation model.** Health as personal value moderating the effect of workplace culture of health on employee engagement.

engagement ($b_3$ = .23, t(167) = 2.83, p < .01; $\Delta R^2$ = .03, F(1,167) = 8.00, p < .01). Thus, health as personal value was a significant moderator of the relationship between workplace culture of health and employee engagement.

To explore the interaction pattern, simple slopes for culture of health predicting employee engagement depending on the level of health as personal value were investigated. Results suggest that for employees who were low in health as personal value (those with scores 1 standard deviation below the mean), the simple slope was b´ = .41 (t(167) = 3.00, p <. 01), while for employees with a mean level of health as personal value the simple slope was b´ =. 65 (t(167) = 6.59, p <. 001). For employees high in health as personal value (those with scores 1 standard deviation above the mean), the simple slope was b´ = .88 (t(167) = 7.51, p <. 001). Hence, the relationship between workplace culture of health and employee engagement was stronger for employees that attribute great value to their health than it was for employees regarding health as less important. These results are in line with our proposed moderation Hypothesis 3.

To finally jointly investigate the hypothesized effects of workplace culture of health, self-rated health and health as personal value on employee engagement, Hayes' PROCESS Macro, model 5, was used. This is a mediation model that allows the direct effect of workplace culture of health on employee engagement to be moderated. Following recommendations by Hayes [61], we again mean-centered the predictor (i.e. culture of health) and the moderator (i.e. health as personal value) prior to analysis to aid in interpretation. Results of this analysis are presented in Fig 4.

The moderated mediation model explained a substantial proportion of variance in employee engagement ($R^2$ = .33, F(4,166) = 20.06, p <. 001). As before, workplace culture of health was positively associated with self-rated health (a = .17, t(169) = 2.32, p < .05), which in turn was positively related to employee engagement (b = .35, t(166) = 3.61, p < .001). Significance of the indirect effect was once more examined using bootstrapping as recommended by Hayes [61]. The analysis yielded a positive and significant indirect effect of workplace culture of health on employee engagement through self-rated health of a x b = .06 with a 95 percent confidence interval from .003 to .134.

Regarding moderation, the model showed a significant interaction between workplace culture of health and health as personal value on employee engagement after controlling for self-rated health ($c_3´$ = .20, t(4,166) = 2.53, p <. 05; $\Delta R^2$ = .03, F(1,166) = 6.42, p <. 05). Hence, the

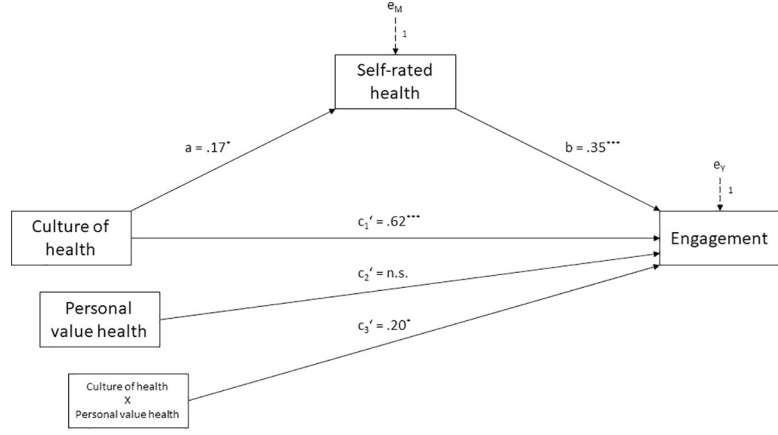

Notes: * p < .05, *** p < .001; n.s. = not significant

**Fig 4. Moderated mediation model.** Health as personal value moderating and self-rated health mediating the effect of workplace culture of health on employee engagement.

direct relationship between workplace culture of health and employee engagement is moderated by health as personal value.

Again, simple slopes were investigated to explore the patterns of conditional direct effects. For employees low in health as personal value (those with scores 1 standard deviation below the mean), the effect of culture of health on employee engagement was $c' = .41$ ($t(166) = 3.10$, p $<. 01$). For employees attributing medium value to health (mean level of health as personal value), the effect of culture of health on employee engagement was $c' = .62$ ($t(166) = 6.47$, p $<. 001$). For employees attributing high value to health (those with scores 1 standard deviation above the mean), the effect of culture of health on employee engagement was $c' = .82$ ($t(166) = 7.15$, p $<. 001$).

Fig 5 shows that the relationship between workplace culture of health and employee engagement was stronger for employees attributing high relevance to health than it was for employees low in health as personal value.

The slope was steeper for employees attributing high value to health, indicating relatively rapid increases in employee engagement with stronger workplace culture of health. On the other hand, the employee engagement slope was relatively lower for employees attributing less value to health. This indicated enhancing interactions, i.e. increasing the moderator increased the effect of workplace culture of health on employee engagement.

We also explored the crossing point of the set of lines depicted in Fig 5, representing the value for culture of health at which health as personal value has no effect on employee engagement, which is $X_{cross\ cent} = .15$ referring to the mean-centered predictor or $X_{cross} = 4.16$ with respect to uncentred workplace culture of health scores. For higher values of culture of health, employees attributing high value to health showed higher engagement than employees attributing less value to health. Likewise, for values of culture of health below this intersection, employees attributing less value to health showed higher engagement than employees attributing high value to health.

To rule out possible confounding effects of the sociodemographic variables gender and age on the associations studied, an additional sensitivity analysis was performed. The aforementioned moderated mediation model was supplemented by gender and age as covariates. Since the inclusion of these potential confounders did not result in any change of the effects reported above by more than 5%, they were not included in the final model.

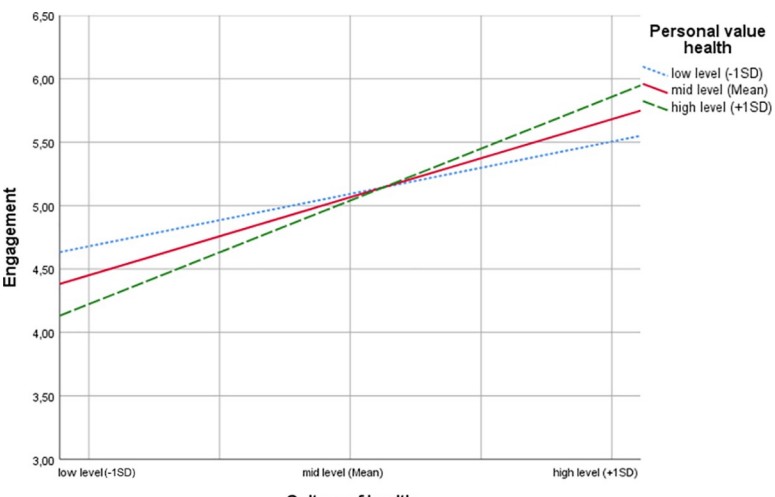

**Fig 5. Interaction plot for employee engagement.**

Taking the aforementioned results together, they supported Hypothesis 1, 2 and 3.

## Discussion

This study examined whether and how a workplace culture of health relates to employee engagement in social enterprises. We conceptualized and empirically tested a moderated mediation model that proposed that a workplace culture of health impacts employee engagement both directly and indirectly through self-rated health, and that the direct relation is moderated by the value employees attribute to health.

### Theoretical implications

The present study advances research on employee engagement. We introduced a workplace culture of health as a novel factor that positively influences employee engagement in organizations. While previous research in the employee engagement domain showed several key antecedents of engagement in terms of specific job characteristics (e.g. task significance or task variety), leadership properties (e.g. transformational leadership), contextual factors (e.g. social support) or personality traits (e.g. conscientiousness, positive affect, optimism) [see, e.g., 22, 66], we add to this knowledge by incorporating an organization's health culture as a contextual antecedent of employee engagement in organizations. Integrating health-related factors not only as potential outcomes of employee engagement (see, e.g., [67]), but also as antecedents, is vital for increasing our understanding about engagement in organizations and advancing theory building in this field, which scholars have called for [12, 22, 66].

Notably, in this regard our investigation also provided insights under which conditions a workplace culture of health is beneficial for employee engagement and thus individual employees as well as social enterprises. We found that employees' personal values in terms of the importance they attribute to health shape the influence a workplace culture of health unfolds on employee engagement. While an organizational culture that prioritizes and promotes employee health and well-being was found to exert positive impact on engagement for employees attributing different value to health, it proved to be particularly influential for employees for whom taking care of health is a high priority in life. Attributing great value to health might help employees to derive positive meaning from a workplace culture of health and to act authentically in everyday work, which contributes to work engagement.

Furthermore, our theoretical considerations and empirical investigation extend research on health promotion in general and workplace culture of health in particular. Prior research in the field of workplace culture of health already demonstrated that such an organizational culture can in principle have positive implications beyond employees' health as it was found, e.g., to impact job satisfaction [57]. Our results on the one hand substantiate this line of reasoning. On the other hand, by providing evidence that a workplace culture of health also influences employee engagement, we expand these elaborations, as engagement is an organizational behavior construct fundamentally different from job satisfaction [12]. Our work suggests that a workplace culture of health is not only related to evaluative judgments employees make about their jobs or job situations, i.e. contentment with status quo and therefore an aspect of satiation, but also to employees' activation.

In addition, by demonstrating that there is a positive direct link between culture of health and engagement as well as an indirect link through employees' self-rated health, this study adds to better understanding the possible mechanisms by which a culture of health unfolds positive motivational effects. The direct link corroborates the reciprocity mechanism proposed, e.g., by Gubler and colleagues [29], positing that employees feel grateful for an organization's health-promotion measures and are thus inclined to reciprocate in terms of

engagement. The established indirect link through self-rated health substantiates the existence of a capability mechanism [29], by which health-promotion measures help employees improving their health and thus their work capability, leading to higher engagement.

Furthermore, our study supports the claim (see, e.g. [68]), that health promotion efforts like establishing a culture of health should not only be investigated with a disease prevention focus, but also in the light of other individual and organizational outcomes. This might have implications for future research on health culture and health promotion.

Finally, this study advances research on the management of social enterprises, which scholars have called for [16, 17], in two important aspects. By emphasizing the importance of a workplace culture of health, it adds a new perspective to the library of works dealing with organizational culture in social enterprises, and it also contributes to research regarding further professionalization of human resource management practices in this type of organizations.

## Practical implications

The present study also has important implications for management practice. The demonstrated positive relation between workplace culture of health and employee engagement implies that managers in social enterprises have one more option to foster engagement and thus improve employees' work performance and organizational citizenship behavior and prevent employee fluctuation. Consequently, we first advise practitioners to become aware of and responsive for this positive link, and to systematically and regularly assess and evaluate the social enterprise's culture with respect to the degree by which employee health and well-being are prioritized and promoted on all cultural levels.

Second, we highly recommend nurturing a workplace culture of health in social enterprises. To do so, crafting a compelling health-related vision, establishing clear health policies and procedures, initiating initiatives and programs to support and improve employee health, encouraging and supporting employees in health matters on a daily basis, acting as a role model regarding healthy behaviors, and allocating adequate resources to the aforementioned endeavors and measures would seem to be beneficial. In doing so, however, managers need to be aware that presumably not all measures promoting a workplace culture of health necessarily also contribute to fostering employee engagement. Health policies like a ban on smoking, e.g., may have a positive effect on employee health [69], but may not have any effect on employee engagement. Thus, measures should be planned carefully with objectives in mind.

Importantly, our finding that the relationship between workplace culture of health and employee engagement is moderated by the importance employees attribute to health does not imply, that social enterprises should only pursue on nurturing a health culture if employees are highly health conscious. The study results indeed suggest that a workplace culture of health is particularly relevant in this case, since employees attributing high value to health exhibit comparatively low engagement if a health culture is weakly developed, while a more pronounced health culture has a strong leverage on engagement. However, based on our empirical findings we also recommend nourishing a workplace culture of health if employees are less health-oriented, since there is a positive association between health culture and employee engagement in this case as well.

Our findings also demonstrated that a workplace culture of health is positively linked to employees' health perceptions. From this point of view, managers in social enterprises should not only strive for fostering a workplace culture of health as a means to increase employee engagement, but also regard it as a lever to prevent or counter health impairments and improve employees' health. Against the background of the challenges social enterprises are facing, nurturing a workplace culture of health would thus offer social enterprises a rational

business advantage since it would quite certainly help to preserve employees' working capacity. At the same time, doing so would be an ethically appropriate approach, as the respective social enterprise would take responsibility for its employees' health.

Taken together, based on this study, better management strategies can be devised to simultaneously improve employees' health and engagement in social enterprises, hitting two birds with one stone.

## Limitations and future research

Like any empirical study, the present investigation is not without shortcomings. A first limitation is associated with the cross-sectional design of our study. This research used workplace culture of health as an explanatory variable and employee engagement as a dependent variable, but cross-sectional data generally allows for reverse causality. Employee engagement could very well have an influence on a workplace culture of health. It would be conceivable, e.g., that highly engaged employees strongly advocate for and work towards establishing a culture of health. Although, based on the theoretical argumentation provided above, the directions of causality implied in this study are likely, we must, therefore, remain cautious in inferring causal, unidirectional relationships. Future research might thus create an even firmer base for the direction of the association between workplace culture of health and employee engagement via longitudinal or experimental study designs.

A second limitation is that all of the study's participants were working for social enterprises located in Germany. Hence, the sample was relatively homogeneous with regard to the general social culture in which the culture of the respective social enterprise was embedded in. The associations identified in this study might present different patterns when investigated in other countries with different cultures and other health-related values. Therefore, scholars could investigate the suggested relationships in other contexts in order to further generalize the current findings.

Third, though we relied on approaches proven in prior research, the measurement of the study constructs could be a potential limitation of this investigation. In operationalizing workplace culture of health, e.g., the present study employed the Workplace Culture of Health Scale developed by Kwon and colleagues [25]. Looking at this scale from the perspective of Schein's [9] theory of organizational culture, the scale seems to particularly focus on the relatively well perceptible artifact level of health culture (such as health promotion programs), while deeper cultural layers of values and assumptions are examined in less detail. While this approach is common considering how other cultural constructs such as an ethical culture [e.g. 70] or a market-oriented culture [e.g. 71, 72] are usually operationalized, relying on measurement instruments of a culture of health that pay more attention to deeper cultural levels could provide additional insights. Since such measurement instruments are not yet available, scholars might embark on developing more refined workplace culture of health scales, e.g. by drawing on research on scales for measuring organizational culture [73]. In a similar vein, this study's reference to self-rated health and thus subjective assessments of health status rather than objective health data could be a potential limitation. We had to refrain from taking such objective data into account for reasons of research economy. Although research in favor of our approach has demonstrated that self-rated health is generally consistent with objective health status [e.g. 74, 75], researchers might validate our findings incorporating objective health data such as fitness level, physical activity level or BMI in future research efforts.

Finally, only gender and age were analyzed as potential confounders. Failure to adequately evaluate factors as potential confounders can bias study results and lead to erroneous conclusions. Hence, it would be an achievement if future studies would consider other possible confounding factors.

Beyond addressing limitations, this study opens up a number of avenues for future research. With regard to health as a personal value this investigation focused on the moderating role of the importance of health for employees. However, it may not only be the case that different employees attach different importance to this value, but that employees also consider various aspects to be desirable when it comes to health issues. While some employees in this regard may, e.g., value disciplined health enhancement, others may value enjoyment and pleasure as sources of health [76, 77]. In other words, there might be various health values and specific preferences at the individual level, and future studies may explore their role in more detail. It could be worth investigating whether a health culture exerts the more influence on employee engagement the more it is fitted towards employees' specific health values. In addition, future studies may investigate in more detail the influence of the various dimensions of a workplace culture of health on employee engagement, also considering their potential interactions.

## Supporting information

**S1 Data. Dataset of the study.**
(SAV)

## Author Contributions

**Conceptualization:** Patrick Nekula, Clemens Koob.

**Data curation:** Patrick Nekula.

**Formal analysis:** Patrick Nekula, Clemens Koob.

**Investigation:** Patrick Nekula, Clemens Koob.

**Methodology:** Patrick Nekula, Clemens Koob.

**Project administration:** Patrick Nekula.

**Supervision:** Clemens Koob.

**Validation:** Patrick Nekula, Clemens Koob.

**Visualization:** Patrick Nekula, Clemens Koob.

**Writing – original draft:** Patrick Nekula, Clemens Koob.

**Writing – review & editing:** Patrick Nekula, Clemens Koob.

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
