## [Decision Letter · Decision Letter 0]

4 Nov 2020

PONE-D-20-22143

Associations between culture of health and employee engagement in social enterprises: A cross-sectional study

PLOS ONE

Dear Dr. Koob,

Thank you for submitting your manuscript to PLOS ONE. After careful consideration, we feel that it has merit but does not fully meet PLOS ONE’s publication criteria as it currently stands. Therefore, we invite you to submit a revised version of the manuscript that addresses the points raised during the review process.

Please consider my comments below around structure and content of information given. 

We look forward to receiving your revised manuscript.

Kind regards,

Andrew Soundy

Academic Editor

PLOS ONE

Journal Requirements:

2. Please amend your list of authors on the manuscript to ensure that each author is linked to an affiliation. Authors’ affiliations should reflect the institution where the work was done (if authors moved subsequently, you can also list the new affiliation stating “current affiliation:….” as necessary).

Additional Editor Comments (if provided):

Introduction

This section is too long and you need to consider how you lead the reader into this area in a succinct way. I quite like the sentences from the abstract in how you lead the reader in so may be think what would a bullet point plan look like.

Line 56-58 just some further explanation as why aging of the workforce present major peril? Can you explain for the reader – you link it to age related illness and diseases – but so what is the problem do they prevent work?

The three paragraphs from line 54-66 could be condensed

Line 69 Can you define cultural anchoring for the reader

Line 80 consider the word proven if statistics are involved detail the stats

Line 90 for me more is needed around what is limited about past work references 17,18

Lines 94-97 – move higher, end the section with your aim – so re-integrate this information above

Within the abstract you mention job demand resource theory but don’t mention it here – think about where it comes and in what order.

Think about this section because your aim to consider associations between workplace culture and health - the reader needs to understand the importance and need for this as a primary focus – so what is known, not known and what further needs to be done should form the major component of the later sections

Looking at the later sections as you lead to your three hypothesis – this needs to be shortened with a consideration of literature which explains and considers past literature. I would like the introduction to end with a consideration of past literature so I would think about the location of introducing the theory aspects.

Think about the outcome measures you use and make sure you consider each of these areas represented by the outcome measure so the reader understands the need to bring this set of outcome measures together

Method

Can you follow a checklist like STROBE for how you present – make sure you consider/address any confounding variables that may influence the results

Introduce healings like sampling, sample size, eligibility criteria and give details within these sections

Results are presented can these be placed in results

Outcomes measures section needs a consideration as to what demographics you obtained and make sure you identify any confounding variables with in this. E.g., if you are considering health should you measure fitness level or physical activity level – surely this will impact on the responses you want? What about BMI?

You are missing your analysis section I see it in the results – but you need more detail regarding your analysis.

For the process of translation of the workplace culture of health scale please included in a supplementary page

Line 331 – when you refer to this work as a survey – for me that is different than what you have done you have done a cross sectional study using validated outcome measures? Please consider this.

Results

Consider presenting this section according to STROBE

Reviewers' comments:

Reviewer's Responses to Questions

**Comments to the Author**

1. Is the manuscript technically sound, and do the data support the conclusions?

Reviewer #1: Yes

2. Has the statistical analysis been performed appropriately and rigorously? 

Reviewer #1: Yes

3. Have the authors made all data underlying the findings in their manuscript fully available?

Reviewer #1: Yes

4. Is the manuscript presented in an intelligible fashion and written in standard English?

Reviewer #1: Yes

5. Review Comments to the Author

Reviewer #1: I am quite glad that I have opportunity to review this paper. It is cleverly designed and produced well fashioned. Actually, while reading it I was under impression that this paper is part of some long and meticulous work (dissertation?) and that authors are describing findings sometimes in a way that is showing their "joy of finding new knowledge". I enjoyed that.

Altogether, my impression is that this is one meticulous paper giving sound answers and opening some important questions in the area of health culture and workplace.

6. PLOS authors have the option to publish the peer review history of their article (what does this mean?). If published, this will include your full peer review and any attached files.

Reviewer #1: **Yes: **Brborovic, Ognjen

---

## [Author Response · Author response to Decision Letter 0]

24 Nov 2020

Clemens Koob (corresponding author) | Catholic University of Applied Sciences Munich

Preysingstraße 95 | 81667 Munich | Germany | clemens.koob@ksh-m.de

November 24, 2020

Dear Dr. Soundy,

we thank you and the reviewer for a thorough reading and constructive criticism of our manuscript entitled “Associations between culture of health and employee engagement in social enterprises: A cross-sectional study” (PONE-D-20-22143) and for the opportunity to revise and resubmit.

In the revised manuscript, we have carefully considered the suggestions and we have edited the manuscript accordingly to address the concerns regarding structure and content of information given. In particular, we have significantly condensed and restructured the introduction, including the later sections leading to the hypotheses, so that the reader better understands the aim of our study, its importance, and the need to bring our outcome measures together. Also, we aligned the method and results sections according to STROBE. 

On the following pages, you will find our responses to the editor and reviewer comments. The responses are coded as follows: 

a) Comments from the editor or reviewer are in italics. 

b) Our responses are shown under each comment in blue and bold.

c) Where beneficial, there are specific references to certain lines in the manuscript in the format “LXXX”; the line numbers refer to the unmarked version of the revised paper without tracked changes (“Manuscript“). 

The inputs were very helpful overall, and we are appreciative of such feedback. 

We look forward to hearing from you regarding our submission. We would be glad to respond to any further questions and comments that you may have.

Sincerely,

Prof. Dr. Clemens Koob

Professor of Management

 

Responses to editor’s comments: 

Journal Requirements:

Level 1, 2 and 3 heading formats and the main text body format were adjusted according to the manuscript body formatting guidelines. 

Figure citations and captions were adjusted to “Fig X”. 

The symbol used to indicate equal contributions of the authors was adjusted to “Pilcrow (paragraph symbol)” according to the title, author, affiliations formatting guidelines.

Corresponding author’s initials were added in parentheses after the email address.

We hope that it now fits the style requirements, as described in the referred templates. 

2. Please amend your list of authors on the manuscript to ensure that each author is linked to an af-filiation. Authors’ affiliations should reflect the institution where the work was done (if authors moved subsequently, you can also list the new affiliation stating “current affiliation:….” as neces-sary).

Thank you for the comment. We added the affiliation of the first author, so that now all authors are correctly linked. 

 

Additional Editor Comments (if provided):

The authors would like to thank the editor for his comments. Care has been taken to improve the work and address the concerns as per the specific comments below.

Introduction

This section is too long and you need to consider how you lead the reader into this area in a succinct way. I quite like the sentences from the abstract in how you lead the reader in so may be think what would a bullet point plan look like.

In response to your concerns about the length, we have substantially shortened the introductory part by approximately 22% (50 lines  39 lines).

With regard to your concerns about the structure, we have adapted the structure according to the line of argumentation in the abstract. 

Line 56-58 just some further explanation as why aging of the workforce present major peril? Can you explain for the reader – you link it to age related illness and diseases – but so what is the problem do they prevent work?

We have expanded the explanation for the reader accordingly (lines 51-54).

The three paragraphs from line 54-66 could be condensed

This text passage has been significantly condensed in the course of the shortening of the entire introductory part.

Line 69 Can you define cultural anchoring for the reader

The corresponding text passage has been crossed out in the course of the streamlining of the introductory part.

Line 80 consider the word proven if statistics are involved detail the stats

Thank you for the comment. We have replaced “proven” with the phrase “was generally found to be effective in prior studies”. Though backing statistics would be available, we have decided to refrain from providing numbers, because from our point of view they would lead away from a succinct introduction at this point.

Line 90 for me more is needed around what is limited about past work references 17,18

We have reformulated this passage in the course of the restructuring of the introductory part. The focus is now on the assessment that there are no studies yet about how a workplace culture of health is related to employee engagement.

Lines 94-97 – move higher, end the section with your aim – so re-integrate this information above

We have re-integrated this information as suggested, so that the section now ends with the aim of the study. 

Within the abstract you mention job demand resource theory but don’t mention it here – think about where it comes and in what order.

Please see the following comment.

Think about this section because your aim to consider associations between workplace culture and health - the reader needs to understand the importance and need for this as a primary focus – so what is known, not known and what further needs to be done should form the major component of the later sections

We agree with the editor’s judgment that the introductory text passages should focus on the importance of investigating the associations between a workplace culture of health and employee en-gagement and on highlighting this aim. Hence and as explained above, we have restructured the first introductory part accordingly.

In addition, we have restructured the further manuscript so that it becomes clearer that (in accordance with the STROBE checklist) the next text section within the introduction outlines the conceptual background for our study (“Theoretical background”). This comprises the job demands-resources theory and the social-ecological workplace culture of health model as the two major building blocks. Accordingly, these two building blocks can be found as subsections, and we consider this is to be the right place to mention and explain the JD-R theory. To be consistent, the social-ecological workplace culture of health model as second theoretical building block is now also mentioned in the abstract (lines 28-29). 

Looking at the later sections as you lead to your three hypothesis – this needs to be shortened with a consideration of literature which explains and considers past literature. I would like the introduction to end with a consideration of past literature so I would think about the location of introducing the theory aspects.

Please see the following comment.

Think about the outcome measures you use and make sure you consider each of these areas repre-sented by the outcome measure so the reader understands the need to bring this set of outcome measures together

According to the STROBE checklist, we have now assigned a separate subsection to the derivation of the hypotheses (“Research hypotheses”) to further clarify the structure for the reader. 

In response to your concerns about the length of this subsection, we have substantially shortened this part by approximately 20% (110 lines  88 lines).

To further clarify the argumentation for the reader, each of the three hypotheses is now specified considering the relevant literature in a separate sub-point, and the respective sub-points have been titled in such a way that the reader better understands how the set of outcome measures relates to each other. In addition, we have also made amendments to the text to support the reader in understanding the need of bringing the different outcome measures together.

Method

Can you follow a checklist like STROBE for how you present – make sure you consider/address any confounding variables that may influence the results

We have now adjusted the presentation of the method according to the STROBE checklist for cross-sectional studies. 

Thank you for the note regarding potential confounders; please take a look at the corresponding comment below in this regard.

Introduce healings like sampling, sample size, eligibility criteria and give details within these sections

We have added subheadings aligned with the STROBE checklist (“Study design and setting”, “Participants”, …), and have provided the respective details within the sections. 

Results are presented can these be placed in results

The results regarding the participants are now presented in the results section (“Participant data”). 

Outcomes measures section needs a consideration as to what demographics you obtained and make sure you identify any confounding variables with in this. E.g., if you are considering health should you measure fitness level or physical activity level – surely this will impact on the responses you want? What about BMI?

We have supplemented the subsection “Measurement of main variables” with information on the two demographic factors that were recorded and investigated as potential confounders (gender, age; s. lines 287-289).

Thank you also for the valuable comment regarding the consideration of more objective health data such as fitness level, physical activity or BMI. Within the scope of this study, we had to refrain from taking such objective data into account for reasons of research economy, and thus had focused on self-rated health and insofar subjective assessments of health status. Although research in favor of our approach has demonstrated that self-rated health is generally consistent with objective health status (see e.g. references 74, 75]), this certainly is a potential limitation; hence, this aspect is addressed in the “Limitations and future research” section, lines 553-559.

You are missing your analysis section I see it in the results – but you need more detail regarding your analysis.

The respective information is now provided within the method section under the subsection “Statistical analyses”. We have also provided further details regarding the analyses. 

For the process of translation of the workplace culture of health scale please included in a supple-mentary page

The workplace culture of health scale can and needs to be licensed from the University of Michigan Health Management Research Center. The license was obtained for this study, with the question set not to be disclosed. From our perspective, the inclusion of a supplementary page describing the translation process would largely amount to a paraphrasing of Beaton et al. (2000) (reference [60]). Hence, instead of including such a page we have provided more details directly in the manuscript (s. lines 279-283).

Line 331 – when you refer to this work as a survey – for me that is different than what you have done you have done a cross sectional study using validated outcome measures? Please consider this.

We have adjusted the formulation accordingly. 

Results

Consider presenting this section according to STROBE

We have adjusted the presentation of the results taking the STROBE checklist for cross-sectional studies into account as following:

• “Participant data” provides details on the study participants (number, characteristics …)

• “Descriptive statistics and correlations” reports means, standard deviations etc. of main study variables

• “Hypothesis testing” reports the main results with regard to our three hypotheses and also provides information on the sensitivity analysis including the potential demographic confounders (lines 428-432).

Responses to reviewer’s comments: 

Reviewers' comments:

Reviewer's Responses to Questions

Comments to the Author

1. Is the manuscript technically sound, and do the data support the conclusions?

The manuscript must describe a technically sound piece of scientific research with data that supports the conclusions. Experiments must have been conducted rigorously, with appropriate controls, repli-cation, and sample sizes. The conclusions must be drawn appropriately based on the data presented.

Reviewer #1: Yes

2. Has the statistical analysis been performed appropriately and rigorously?

Reviewer #1: Yes

3. Have the authors made all data underlying the findings in their manuscript fully available?

The PLOS Data policy requires authors to make all data underlying the findings described in their manuscript fully available without restriction, with rare exception (please refer to the Data Availabil-ity Statement in the manuscript PDF file). The data should be provided as part of the manuscript or its supporting information, or deposited to a public repository. For example, in addition to summary statistics, the data points behind means, medians and variance measures should be available. If there are restrictions on publicly sharing data—e.g. participant privacy or use of data from a third party—those must be specified.

Reviewer #1: Yes

4. Is the manuscript presented in an intelligible fashion and written in standard English?

Reviewer #1: Yes

5. Review Comments to the Author

Reviewer #1: I am quite glad that I have opportunity to review this paper. It is cleverly designed and produced well fashioned. Actually, while reading it I was under impression that this paper is part of some long and meticulous work (dissertation?) and that authors are describing findings sometimes in a way that is showing their "joy of finding new knowledge". I enjoyed that.

Altogether, my impression is that this is one meticulous paper giving sound answers and opening some important questions in the area of health culture and workplace.

The authors would like to thank the reviewer for these comments.

6. PLOS authors have the option to publish the peer review history of their article (what does this mean?). If published, this will include your full peer review and any attached files.

Do you want your identity to be public for this peer review? For information about this choice, in-cluding consent withdrawal, please see our Privacy Policy.

Reviewer #1: Yes: Brborovic, Ognjen

While revising your submission, please upload your figure files to the Preflight Analysis and Conver-sion Engine (PACE) digital diagnostic tool, https://pacev2.apexcovantage.com/. PACE helps ensure that figures meet PLOS requirements. To use PACE, you must first register as a user. Registration is free. Then, login and navigate to the UPLOAD tab, where you will find detailed instructions on how to use the tool. If you encounter any issues or have any questions when using PACE, please email PLOS at figures@plos.org. Please note that Supporting Information files do not need this step.

In compliance with data protection regulations, you may request that we remove your personal reg-istration details at any time. (Remove my information/details). Please contact the publication office if you have any questions.

-----

Thank you again to the editor and the reviewer for the time and effort.

---

## [Decision Letter · Decision Letter 1]

28 Dec 2020

Associations between culture of health and employee engagement in social enterprises: A cross-sectional study

PONE-D-20-22143R1

Dear Dr. Koob,

We’re pleased to inform you that your manuscript has been judged scientifically suitable for publication and will be formally accepted for publication once it meets all outstanding technical requirements.

Kind regards,

Andrew Soundy

Academic Editor

PLOS ONE

Additional Editor Comments (optional):

Reviewers' comments:

Reviewer's Responses to Questions

**Comments to the Author**

1. If the authors have adequately addressed your comments raised in a previous round of review and you feel that this manuscript is now acceptable for publication, you may indicate that here to bypass the “Comments to the Author” section, enter your conflict of interest statement in the “Confidential to Editor” section, and submit your "Accept" recommendation.

Reviewer #1: All comments have been addressed

2. Is the manuscript technically sound, and do the data support the conclusions?

Reviewer #1: Yes

3. Has the statistical analysis been performed appropriately and rigorously? 

Reviewer #1: Yes

4. Have the authors made all data underlying the findings in their manuscript fully available?

Reviewer #1: Yes

5. Is the manuscript presented in an intelligible fashion and written in standard English?

Reviewer #1: Yes

6. Review Comments to the Author

Reviewer #1: (No Response)

7. PLOS authors have the option to publish the peer review history of their article (what does this mean?). If published, this will include your full peer review and any attached files.

Reviewer #1: No

---

## [Editor Report · Acceptance letter]

8 Jan 2021

PONE-D-20-22143R1 

Associations between culture of health and employee engagement in social enterprises: A cross-sectional study 

Dear Dr. Koob:

I'm pleased to inform you that your manuscript has been deemed suitable for publication in PLOS ONE. Congratulations! Your manuscript is now with our production department. 

Kind regards, 

on behalf of

Dr. Andrew Soundy 

Academic Editor

PLOS ONE